# Molten Chlorides as the Precursors to Modify the Ionic Composition and Properties of LiNbO_3_ Single Crystal and Fine Powders

**DOI:** 10.3390/ma15103551

**Published:** 2022-05-16

**Authors:** Nikolay A. Viugin, Vladimir A. Khokhlov, Irina D. Zakiryanova, Vasiliy N. Dokutovich, Boris D. Antonov

**Affiliations:** Institute of High Temperature Electrochemistry of the Ural Branch of the Russian Academy of Sciences, 620066 Ekaterinburg, Russia; optica96@ihte.uran.ru (I.D.Z.); v.dokutovich@ihte.uran.ru (V.N.D.); b.antonov@ihte.uran.ru (B.D.A.)

**Keywords:** lithium metaniobate, molten salt, ionic composition, modifying, structure, morphology, XRD diffractometry, Raman, IR spectroscopy

## Abstract

Modifying lithium niobate cation composition improves not only the functional properties of the acousto- and optoelectronic materials as well as ferroelectrics but elevates the protonic transfer in LiNbO_3_-based electrolytes of the solid oxide electrochemical devices. Molten chlorides and other thermally stable salts are not considered practically as the precursors to synthesize and modify oxide compounds. This article presents and discusses the results of an experimental study of the full or partial heterovalent substitution of lithium ion in nanosized LiNbO_3_ powders and in the surface layer of LiNbO_3_ single crystal using molten salt mixtures containing calcium, lead, and rare-earth metals (REM) chlorides as the precursors. The special features of heterovalent ion exchange in chloride melts are revealed such as hetero-epitaxial cation exchange at the interface PbCl_2_-containing melt/lithium niobate single crystal; the formation of ${\mathrm{Li}}_{\left(1-x\right)\;}{\mathrm{Ca}}_{\left(x/2\right)}V_{\left(x/2\right)}^{{\mathrm{Li}}^{+}}{\;\mathrm{NbO}}_{3}$ solid solutions with cation vacancies as an intermediate product of the reaction of heterovalent substitution of lithium ion by calcium in LiNbO_3_ powders; the formation of lanthanide orthoniobates with a tetragonal crystal structure such as scheelite as the result of lithium niobate interaction with trichlorides of rare-earth elements. It is shown that the fundamental properties of ion-modifiers (ion radius, nominal charge), temperature, and duration of isothermal treatment determine the products’ chemical composition and the rate of heterovalent substitution of Li^+^-ion in lithium niobate.

## 1. Introduction

Niobates of alkali, alkaline-earth, rare-earth, and transition metals are practically important acousto- and optoelectronic materials, ferroelectrics. Therefore, the development of new methods to obtain them, especially in the form of nanoparticles or thin films, is of interest to create materials with new functional possibilities for special devices with high service properties. Due to the unique optical and dielectric properties, these compounds can be used to create new optoelectronic devices and solid oxide electrochemical devices [1,2,3,4,5,6,7,8,9,10,11]. It is possible to use nanostructured LiNbO_3_ doped with rare earth ions as self-doubling laser materials [12]. Ligating additives significantly affect the optical properties of the LiNbO_3_ crystal, for example, the change in its photorefractive and electrooptic properties [13,14,15]. Fe or Ce doping is followed by a photoconductivity increase and a consequent decrease of photorefraction [16,17]. It is shown that the ion impurities of different metals (Mg, Sc, Ce, Cu, Zn, In, Fe) in a LiNbO_3_ single crystal suppress the photorefractive properties and elevate the electrooptic properties [18]. The possibility of fabrication of optical waveguides from LiNbO_3_ doped with transition metals was noted in [19,20,21]. It is promising to use the LiNbO_3_ and PbNb_2_O_6_ piezoelectric properties to create novel ultrasonic devices for medical diagnostics [22,23,24,25,26,27,28].

In recent years, more advanced molten salt methods for the synthesis of niobates of alkali, alkaline earth, rare earth, and transition metals in the shape of nanopowders with different particle morphology have been proposed and tested [29,30,31,32]. These methods enable to decrease the process temperature by hundreds of degrees in contrast to solid-phase synthesis, to ensure uniform distribution of the precursor particles in the reaction medium, to increase the rate of the synthesis, to control the morphology and size of reaction products, and to reduce their tendency to agglomeration. Practically in all studies at different stages of the preparation of niobates, thermally unstable compounds are used as precursors. In some cases, thermally stable molten alkali chlorides or their low-melting mixtures are applied as the auxiliary substances to improve synthesis procedure [29]. Judging by the publications, these and other thermally stable salts were not considered as the precursors to synthesize complex oxides. A number of our works [31,32] showed that the use of halide melts as precursors create more suitable conditions for the synthesis of nanosized particles. The logical development of these studies is the use of heat-resistant chloride melts to change the chemical composition of complex oxide powders and obtain thin films on single-crystal surface.

This article presents and discusses the results of an experimental study of the heterovalent substitution of lithium ion in nanosized LiNbO_3_ powders and in the surface layer of LiNbO_3_ single crystal using molten salt mixture containing calcium, lead, and rare-earth metal (REM) chlorides as the precursors.

## 2. Materials and Methods

### 2.1. Initial Materials and Molten Salt Reaction Media

Modifying of the ionic composition was performed for the single crystalline and nanosized powder samples of lithium niobate. The plates with dimensions of 60 × 18 × 2 mm were cut from the LiNbO_3_ single crystal along the main optical axis in the YZ plane (LLC “Quant”, St. Petersburg, Russia).

A new molten salt synthesis method was used for obtaining initial low-sized lithium niobate powders [32]. It is based on the reaction between the lithium oxide and niobium pentoxide formed as the result of the interactions of molten lithium chloride and niobium pentachloride with air oxygen:2LiCl(l) + 2NbCl_5_(l) + 3O_2_(g) = 2LiNbO_3_(s) + 6Cl_2_(g), (ΔG = −525.1 kJ/mol at 700 °C).(1)

A simplified version of this method with the low-sized Nb_2_O_5_ (JSC “Vekton”, Russia, St. Petersburg) as precursor was also used:2LiBr(l) + Nb_2_O_5_(l) + 0.5O_2_(g) = 2LiNbO_3_(s) + Br_2_(g) (ΔG = −130.3 kJ/mol at 700 °C).(2)

In either case the one-phase lithium niobate (Figure 1) was obtained with mean powder particle size equal to about 250 nm (Figure 2). The heterovalent substitution of lithium ions in LiNbO_3_ was performed in molten salt reaction media which were the binary mixtures of precursors: calcium dichloride, lead dichloride, and lanthanide trichlorides with the alkali metal salts (LiCl, NaCl, KCl, and KNO_3_) to provide a substantial decrease of operating temperature.

Anhydrous CaCl_2_ and PbCl_2_, pure for analysis, were used to prepare reaction melts. Anhydrous lanthanide trichlorides (CeCl_3_, GdCl_3_, YbCl_3_) were synthesized ordinarily through the chlorination of high purity CeO_2_, Gd_2_O_3_, Yb_2_O_3_ powders with CCl_4_ [33]. A small amount of each of these salts free from foreign impurities (moisture and hydrocarbons) was melted together with dehydrated lithium chloride for subsequent use as a reaction medium. The composition (in mole fractions) of working reaction salt mixtures prepared in this way and their liquidus temperature (T_m_) are as follows: 0.025PbCl_2_–0.975KNO_3_ (~325 °C); 0.35CaCl_2_–0.65LiCl (470 °C); 0.4CaCl_2_–0.6KCl (717 °C), 0.02LnCl_3_–0.98LiCl (~590 °C).

### 2.2. Technique of Ionic Modifying Lithium Niobate

A simple method of the exposition of the initial LiNbO_3_ samples (single-crystalline plate and low-sized powders) in the above-mentioned reaction melts at the constant temperature was used for the modifying cationic composition of the lithium niobate. The experiments were carried out in the reactors shown schematically in Figure 3. The *b* reactor version allowed the runs under different gas atmospheres (air and argon) at the same conditions to study the possible effect of the air oxygen-containing components on the chemical composition of the reaction products.

The holding temperature depended on the reaction medium melting point. The LiNbO_3_ single-crystalline plate was treated in PbCl_2_–KNO_3_ melt at 360 °C to eliminate potassium nitrate thermal decomposition. Lithium niobate powders were processed in chloride melts containing CaCl_2_, CeCl_3_, GdCl_3_, and YbCl_3_ at 700 °C or 750 °C. The isothermal holding time had been no less than 5 h.

All preliminary operations and experiments with melts containing cerium, gadolinium, and ytterbium trichloride were performed only in an inert gas atmosphere. The preparation of the reaction mixtures and their loading into the reactor were carried out in a dry glove box in an atmosphere of pure nitrogen. After prolonged vacuum treatment of salt mixtures to a temperature of 500 °C, the reactor was filled with high-purity gaseous argon. It allows us to prevent possible undesirable side reactions of rare-earth trichlorides with atmospheric oxygen and moisture, which lead to the formation of oxychlorides that do not participate in heterovalent substitution reactions of lithium in its niobate due to the firm Ln-O bond in these compounds.

After experiments, the LiNbO_3_ single-crystalline plates and oxide-salt mixtures were cooled to room temperature and thoroughly washed with distilled water. The film formed on the single crystal surface and the powdery reaction products filtered were dried and, subsequently, examined in detail.

### 2.3. Examination Methods of the Reaction Products

Various methods were used to determine composition and study the structural, morphological, granulometric, and optical properties of the initial single crystalline and powdery lithium niobate and reaction products. X-ray diffraction analysis was conducted with a D/MAX-2200VL/PC automated X-ray diffractometer (Rigaku Corp., Tokyo, Japan) with the CuKα1 radiation source and a graphite monochromator. The diffractograms were identified using the PDF-2 database. The elemental compositions of the reaction media were researched with an Optima 4300 DS (Perkin Elmer Inc., Wellesley, MA, USA) emission spectrometer. The microscopic structure of the films and powders was studied by the Raman light scattering method using an Ava-Raman fiber-optic spectrometer (Avantes, Eerbeek, The Nederlands) (light source with 532 nm wavelength) as well as by studying the IR spectra using the TENSOR 27 spectrometer (Bruker Optik GmbH, Ettlingen, Germany). The powders’ granulometric composition was determined using a Malvern Instruments Mastersizer 2000 laser diffraction analyser (Malvern Instruments Ltd., Malvern, UK). The semi-quantitative chemical analysis of the reaction products was carried out with a JSM-5900LV scanning electron microscope (Jeol Ltd., Tokyo, Japan) combined with an X-act ADD energy dispersive X-ray detector (Oxford Instruments, Abingdon, UK).

## 3. Results and Discussion

### 3.1. Formation of the Lead Metaniobate Film on the Y-Z Face of LiNbO_3_ Single Crystal

Lead metaniobate should be easily incorporated into the surface of a lithium niobate single crystal with the formation of thin films due to the identity of their rhombohedral structure at temperatures below the Curie point of PbNb_2_O_6_ [34]. After holding LiNbO_3_ single-crystalline plate in PbCl_2_–KNO_3_ melt at 360 °C for 6 h, cooling, washing in distilled water and drying, it was studied in detail with the methods mentioned in Section 2.3.

A slight improvement in the measurement XRD technique was made to obtain a clearer diffraction pattern of changes in the surface layer of the single crystal treated with the salt melt. For this, we used the X-ray grazing incidence diffraction (GID) technique [35]. The diffraction was recorded from a single-crystal surface turned to the radiation source by 1.5° angle. The view of the X-ray diffraction pattern obtained (Figure 4) is similar to the diffractograms observed for ultrathin films formed on the surface of oxide single crystals [36]. The most intensive reflex of the rhombohedral lead metaniobate at 29° ascribed to the crystallographic plane 300 [35,37,38,39,40] is clearly seen in the film X-ray diffractogram.

This was also evidenced by the results of Raman spectroscopy (Figure 5, Table 1), elemental analysis of the reaction medium and the crystal surface after the experiment (Table 2 and Table 3).

The Raman spectrum of LiNbO_3_ crystals is sensitive to changes in the crystal composition upon the introduction of ligating additives [41,42]. It can be used to prove the inclusion of Pb in the crystal lattice.

In the Raman spectrum of single-crystal LiNbO_3_, which has the space symmetry group R3c (Z = 2) [38], vibrational frequencies should be observed, characterizing the longitudinal (LO) and transverse (TO) displacements of ions relative to the main optical axis of the crystal [41]. In the recorded spectrum of the initial LiNbO_3_ single crystal in the backscattering geometry *z* |*xx*| *z* (Figure 5a, curve 1), the observed set of frequencies is in good agreement with [43]. With this orientation of the crystal, both LO and TO vibrations, and it is convenient to compare the changes in the spectral parameters of the original and modified samples (Figure 5a). For a detailed analysis of the spectra, we used their decomposition into Gaussian components (Figure 5b). The decomposition results are shown in Table 1. An intense band at 874 cm^−1^, which does not overlap with other bands, was used as a reference for calculating the normalized intensities in the obtained spectra.

Some increase in the normalized intensity of the bands is marked in the spectrum of the modified sample. The most significant increase in the intensity of the spectral lines is observed at 237, 269, and 620 cm^–1^. It is interesting to note that these bands are the most intensive in the Raman spectrum of LiNbO_3_ powder [32,44].

In addition, in the region of oscillations of oxygen octahedra [NbO_6_], a high-frequency shift of the band is noted from 649 cm^−1^ in the initial sample up to 659 cm^−1^ in the modified one, which indicates an increase in the force constant Nb-O bonds when replacing lithium ions by lead ones with a larger ionic radius.

It would be interesting to compare the Raman spectrum of modified niobate lithium with the spectra of crystalline lead niobate. Unfortunately, we did not find in the literature of systematic studies of the Raman spectra of single crystals or polycrystals of lead niobates. There are isolated data on the Raman frequencies of the Pb_n_Nb_2_O_5+n_ composition [45], for which, with n close to unity, in the Raman spectrum, wide bands are recorded in regions 240–280 and 550–700 cm^−1^. In paper [46], the Raman spectrum of the PbNb_2_O_6_ powder of the orthorhombic modification is given. Wide bands were recorded in the regions of 250–350 and 470–680 cm^−1^. In the IR spectrum of the orthorhombic PbNb_2_O_6_ the bands at 692, 651, 549 cm^−1^ were marked [37], while for the rhombohedral modification of the lead niobate-655, 556 cm^−1^. Note that the IR spectrum was studied in the spectral range higher than 450 cm^−1^. The low-frequency range was not investigated.

In general, the change in the normalized intensities and the high-frequency shift of the vibrational bands in the modified sample as compared to the initial one indicates a deviation of the chemical composition of the surface layers of the LiNbO_3_ sample from stoichiometry due to a change in the [Li]/[Nb] ratio, disordering and deformation of the oxygen octahedra [NbO_6_] and the formation of impurity substitutional defects upon the introduction of lead cations into the crystal lattice of a lithium niobate single crystal.

Elemental compositions of the molten salt reaction medium (Table 2) and the single crystal surface (Table 3) after the experiment have become the additional direct data demonstrating the formation of the PbNb_2_O_6_ film on the LiNbO_3_ single crystal surface.

The results of the X-ray diffraction study, Raman spectroscopy, and elemental analysis of the reaction medium and the crystal surface after the experiment made it possible to conclude that the exchange reaction:2LiNbO_3_(s) + Pb^2+^(m) → PbNb_2_O_6_ (s) + 2Li^+^(m)(3)
proceeds in the surface layer of a LiNbO_3_ single crystal. In general terms, this reaction can be considered as the hetero-epitaxial cation exchange at the molten salt/solid interfacial [47,48].

### 3.2. Modifying Composition of the LiNbO_3_ Fine Powders in Calcium-Containing Chloride Melts

The possibility of the isomorphic heterovalent substitution of lithium by calcium was confirmed by calculation of the reaction Gibbs energy (ΔG):2LiNbO_3_(s) + CaCl_2_(l) = CaNb_2_O_6_(s) + 2LiCl(l)(4)
with the database of the Outokumpu HSC Chemistry 7.1–Software. The ΔG values equal to −9.65 and –14.56 kJ/mol at 700 and 750 °C, respectively, show the reaction (4) must occur. We did not find the thermodynamic data for calcium pyroniobate (Ca_2_Nb_2_O_7_) that did not allow us to evaluate the probability of its formation by the similar reaction.

The modification of the cationic composition of the fine lithium niobate powders in calcium-containing chloride melts was performed in the reactor shown in Figure 3b. Molten mixtures 0.35CaCl_2_–0.65LiCl and 0.40CaCl_2_–0.60KCl with approximately the same molar content of calcium chloride were chosen as modifying reaction mixtures.

The special experiments on the holding LiNbO_3_ powders in a molten LiCl–KCl eutectic at 700 °C during 5 h showed that lithium metaniobate does not react with the melt. No other substances other than the original lithium metaniobate were detected after its isothermal holding in this salt melt. The X-ray diffraction pattern was identical to that shown in Figure 1.

Initially, experiments with 0.35CaCl_2_–0.65LiCl and 0.40CaCl_2_–0.60KCl melts were carried out at 700 °C and 750 °C, respectively, in both argon and air atmospheres. It was found that the nature of the gaseous atmosphere above the reaction mixture did not affect the chemical composition of the reaction products of heterovalent substitution of lithium with calcium. Subsequently, the experiments with LiNbO_3_ modifying in chloride melts were performed under an air atmosphere. It was expected that the contact of the LiNbO_3_ microcrystal powders with these melts would result in the isomorphic heterovalent substitution of lithium ions by calcium ions with the calcium metaniobate (CaNb_2_O_6_) formation since the oxygen affinity of calcium is higher than that of lithium.

The typical X-ray diffraction pattern of the reaction products is shown in Figure 6 where data are presented on the lithium niobate powder held in the 0.35CaCl_2_–0.65LiCl melt for 5 h. It is seen that the reaction of isomorphic heterovalent substitution results in the formation of the bipyramidal CaNb_2_O_6_ with an orthorhombic fersmite-type structure crystallizing in the space group Pbcn (the lattice parameters are: a = 14.926± 0.004 Å b = 5.752 ± 0.004 Å; and c = 5.204 ± 0.004 Å) [49,50].

A comparative analysis of the peaks of initial lithium niobate observed in this diffractogram and ones of the XRD pattern of pure LiNbO_3_ (Figure 1) showed the crystal lattice parameters and unit cell volume increase of the modified powder (a = 5.19630 Å, c = 13.95273 Å, V = 326.27 Å^3^) compared to the initial niobate (a = 5.15252 Å, c = 13.87072 Å, V = 318.91 Å^3^). This indicates the formation of a solid substitution solution with cationic vacancies ${\mathrm{Li}}_{\left(1-x\right)\;}{\mathrm{Ca}}_{\left(x/2\right)}V_{\left(x/2\right)}^{{\mathrm{Li}}^{+}}{\mathrm{NbO}}_{3}$ as an intermediate in the reaction (4).

The typical example considered above, illustrating a simple method for modifying the chemical composition of submicron lithium niobate powders, describes the results of partial replacement of lithium ions in LiNbO_3_ crystal lattice in a relatively short time of powder contact with the salt melt-modifier (4–5 h) without forced mixing heterogeneous oxide-salt mixture. When increased in the duration of the lithium niobate isothermal holding in these melts, the time-controlled reaction could be realized up to the complete substitution of Li^+^-ions by Ca^2+^-ions. As illustrations, Figure 7 and Figure 8 show X-ray diffraction patterns and Raman spectra of the products of interaction of lithium niobate with melts (0.35CaCl_2_–0.65LiCl) and (0.40CaCl_2_–0.60KCl) for 7 h. It can be seen that lithium is fully substituted by calcium in the LiNbO_3_ powder. The only reaction product is stoichiometric calcium metaniobate [50,51,52].

The modification of the chemical composition of nanocrystalline lithium niobate powders in the melts, based on the reaction of isomorphic heterovalent substitution, showed that its products are calcium metaniobate (CaNb_2_O_6_), regardless of the composition of the modifier melt used in the work and the gaseous medium over the oxide-chloride reaction medium. The results obtained demonstrate the efficiency of the proposed method for complete or partial replacement of lithium ions with calcium ones in the LiNbO_3_ crystal lattice.

### 3.3. Modifying Composition of the LiNbO_3_ Fine Powders in Chloride Melts Containing Rare-Earth Ions

More complex processes occur when lithium metaniobate interacts with rare-earth trichlorides. The composition and structure of the products of the reaction of heterovalent replacement of lithium ions with Ln^3+^ ions were determined by the XRD as well as IR and Raman spectroscopy.

According to XRD data, the products of the interaction of lithium niobate with molten mixtures of LiCl-LnCl_3_ (Ln = Ce, Gd, Yb) mainly consisted of REM orthoniobates (LnNbO_4_), LiNb_3_O_8_, niobium oxide Nb_2_O_5_, and residues of the starting material. Typical examples of X-ray diffraction patterns are shown in Figure 9. In the Raman spectra shown in Figure 10, Ag peaks were observed near 180, 330, and 810 cm^−1^, corresponding to symmetric vibrations of the Nb-O bonds in rare-earth metal orthoniobates with a monoclinic fergusonite structure belonging to the space group I2/a. As can be seen from Figure 11, deep minima are observed in the infrared spectra about 470, 670, and 810 cm^−1^, which are characteristic of rare-earth metal orthoniobates. Features of vibrational spectra of products of heterophase substitution of lithium-ion in metaniobate by ions of rare earth metals are consistent with recent literature data [44,53,54]. 

Based on the results obtained, the following equations of the ongoing chemical processes are proposed:4LiNbO_3_ + LnCl_3_ → LnNbO_4_ + LiNb_3_O_8_ + 3LiCl;(5)
3LiNbO_3_ + LnCl_3_ → LnNbO_4_ + Nb_2_O_5_ + 3LiCl.(6)

It is obvious that these processes are interconnected due to the possibility of a reversible solid-phase reaction occurring in the salt melt at a sufficiently low temperature (700 °C):LiNbO_3_ + Nb_2_O_5_ ↔ LiNb_3_O_8_.(7)

It is important to note that, despite the low concentration of rare earth elements in the melt, the latter in all experiments were in excess relative to the stoichiometric ratio of the components in Equations (5) and (6). Taking this into account, it can be concluded that phases with a higher content of rare earth elements cannot be formed under these conditions. The synthesized orthoniobates of rare earth metals (LnNbO_4_) have a fergusonite structure similar to the scheelite structure, which is characterized by a tetrahedral oxygen environment of the Nb atom, in contrast to the starting materials LiNbO_3_ with an octahedral environment. Such essential change in the structure indirectly indicates that a significant contribution to the reaction mechanism, in this case, will be made by the processes of dissolution-precipitation and diffusion in the liquid-salt reaction medium.

A noticeable decrease in the amount of the LnNbO_4_ formed during the reaction was observed ongoing from the Ce-containing reaction mixture to the Yb-containing one, although all experiments were carried out under the same conditions and time. This indicates a decrease in the rate of the heterovalent substitution reaction due to the strengthening of the Ln-Cl bond in REM trichlorides as the radius of Ln^3+^ ions in the Ce-Gd-Yb series decreases.

## 4. Conclusions

This comprehensive targeted research was carried out to search and develop a new unconventional method for functional materials synthesis.

A new methodological approach has been developed to modify the ion composition of lithium metaniobate single crystal and powders in thermally stable salt melts containing lead, calcium, and rare-earth chlorides as the precursors at temperatures not exceeding 800 °C without additional heat treatment (annealing) of the reaction products.

It is shown that the products’ chemical composition and the rate of heterophase reactions depend on the fundamental properties of ion-modifiers (ion radius, nominal charge), temperature, and the duration isothermal treatment of LiNbO_3_ nanocrystalline powders in salt melts.

New information has been obtained on the features of heterovalent ion exchange between LiNbO_3_ and chloride precursors. The following results deserve special attention:hetero-epitaxial cation exchange at the interface PbCl_2_-containing melt/single crystal of lithium niobate;the formation of ${\mathrm{Li}}_{\left(1-x\right)\;}{\mathrm{Ca}}_{\left(x/2\right)}V_{\left(x/2\right)}^{{\mathrm{Li}}^{+}}{\mathrm{NbO}}_{3}$ solid solution with cation vacancies as an intermediate product of the reaction of heterovalent substitution of lithium ion by calcium one in nanocrystalline LiNbO_3_ powders;the formation of tetragonal cerium, gadolinium, and ytterbium orthoniobates with crystal structures like scheelite one as the result of the interaction of the lithium metaniobate with the rare-earth trichlorides.

## Figures and Tables

**Figure 1 materials-15-03551-f001:**
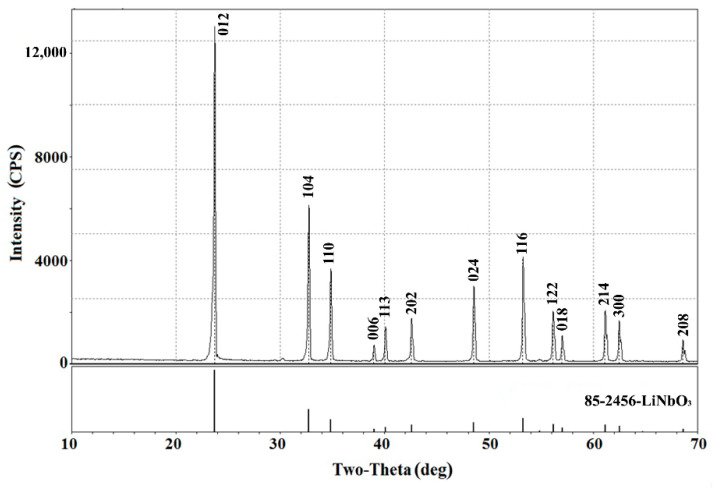
X-ray diffractogram of powder lithium niobate synthesized in molten lithium chloride at 700 °C used in experiments on the modification of their cationic composition.

**Figure 2 materials-15-03551-f002:**
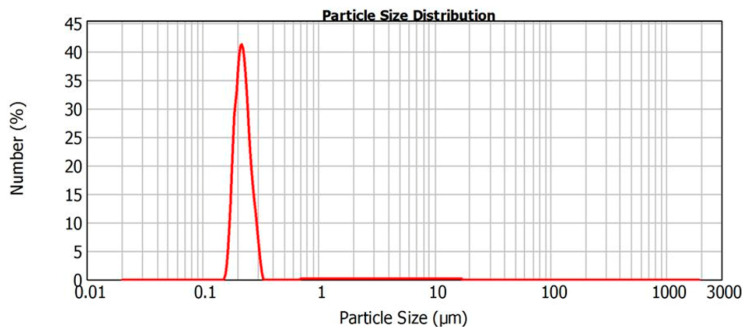
Particle size distribution of synthesized LiNbO_3_ powder.

**Figure 3 materials-15-03551-f003:**
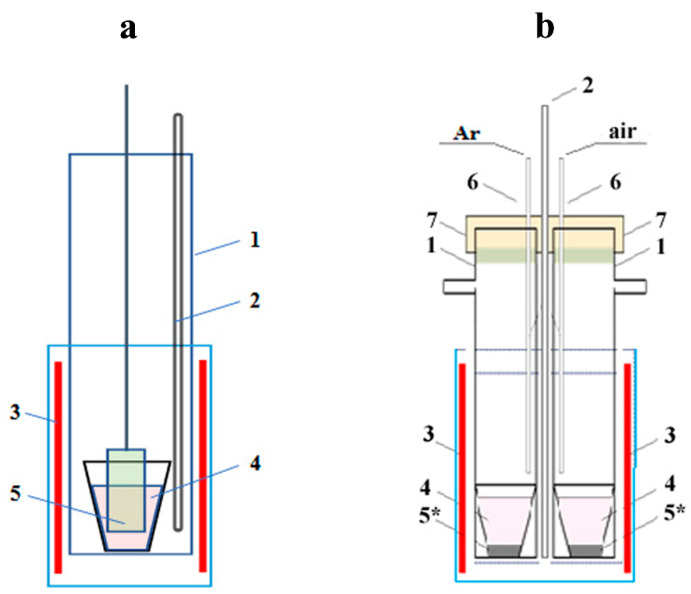
Design of the reactors for the modifying chemical composition of single crystalline (**a**) and powder (**b**) lithium niobate in salt melts: (1) quartz tube, (2) thermocouple, (3) electrical furnace, (4) salt melt, (5) single crystalline plate, (5*) LiNbO_3_ powder, (6) tube for gas supply, (7) stopper.

**Figure 4 materials-15-03551-f004:**
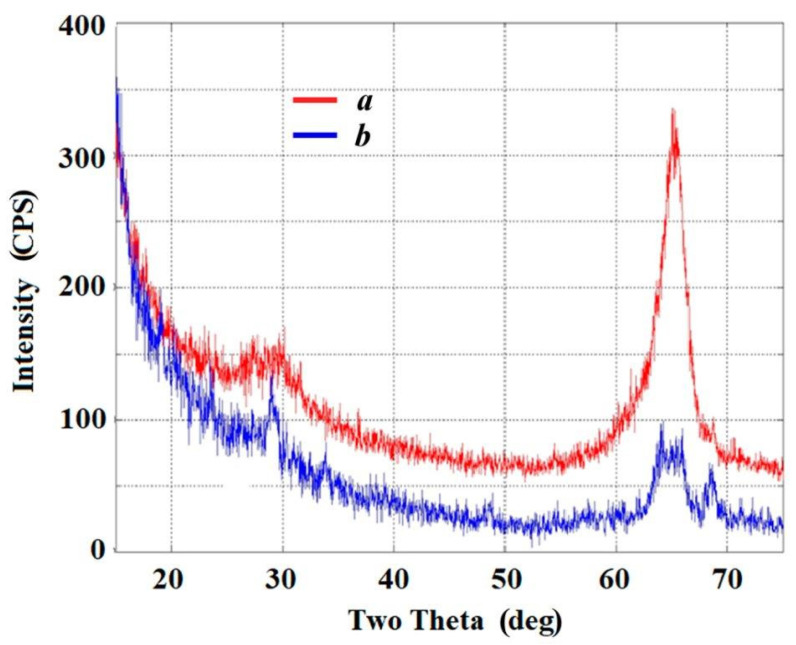
X-ray diffractograms of the LiNbO_3_ single-crystal surface: initial (*a*) and modified (*b*) samples.

**Figure 5 materials-15-03551-f005:**
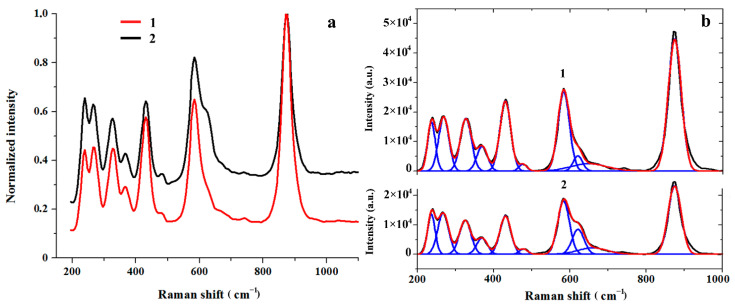
Normalized Raman spectra (1) and their Gaussian expansions (2) of the LiNbO_3_ single crystal surface (*z* |*xx*| *z* scattering geometry): initial (**a**) and modified (**b**) samples.

**Figure 6 materials-15-03551-f006:**
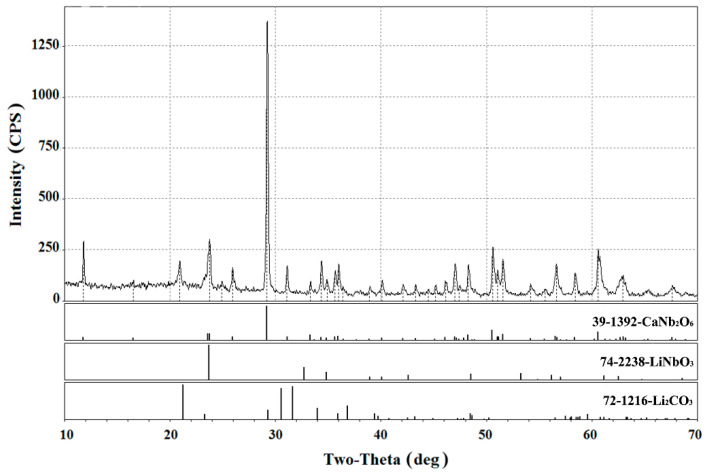
X-ray diffractogram of the heterovalent ionic exchange reaction products after holding lithium niobate in a molten mixture of lithium and calcium chlorides at 700 °C for 5 h.

**Figure 7 materials-15-03551-f007:**
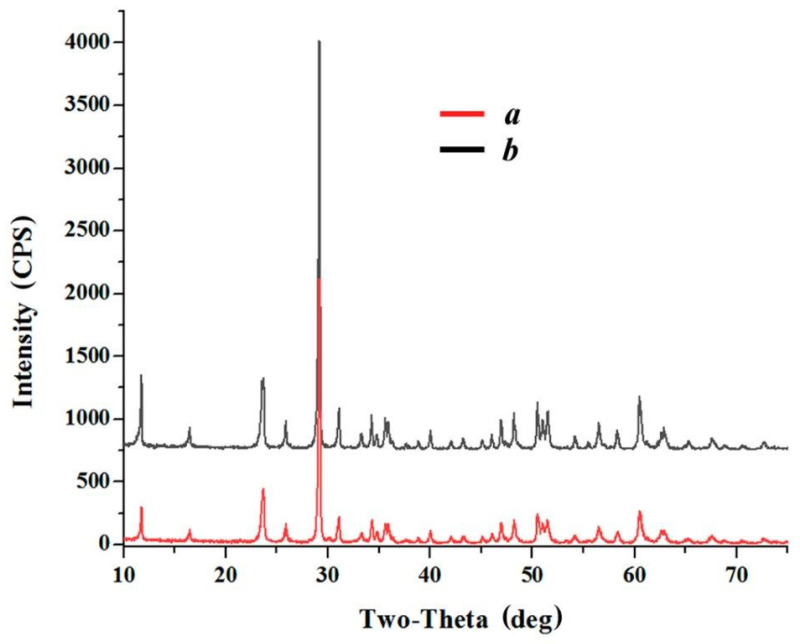
X-ray diffractograms of the calcium metaniobate obtained after holding LiNbO_3_ powders in the melts of (0.40CaCl_2_–0.60KCl) at 750 °C (*a*) and (0.35CaCl_2_–0.65LiCl) at 700 °C (*b*) under air atmosphere for 7 h.

**Figure 8 materials-15-03551-f008:**
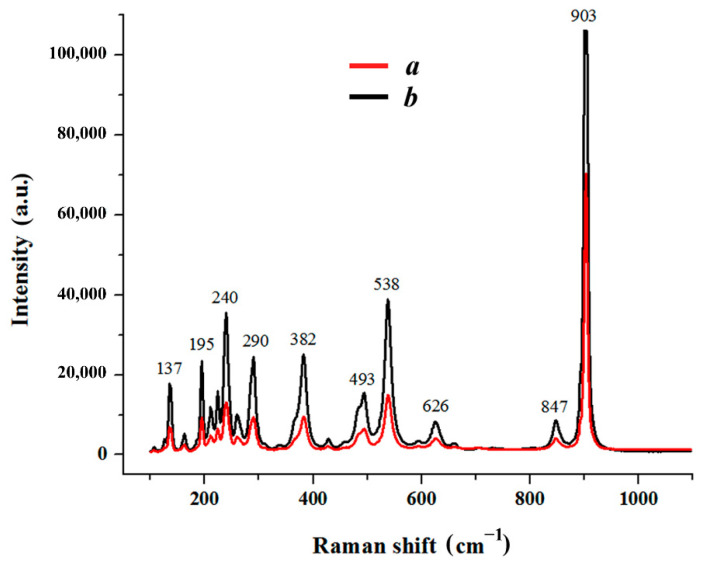
Raman spectra of the calcium metaniobate obtained after holding LiNbO_3_ powders in the melts of (0.40CaCl_2_–0.60KCl) at 750 °C (*a*) and (0.35CaCl_2_–0.65LiCl) at 700 °C (*b*) under air atmosphere for 7 h.

**Figure 9 materials-15-03551-f009:**
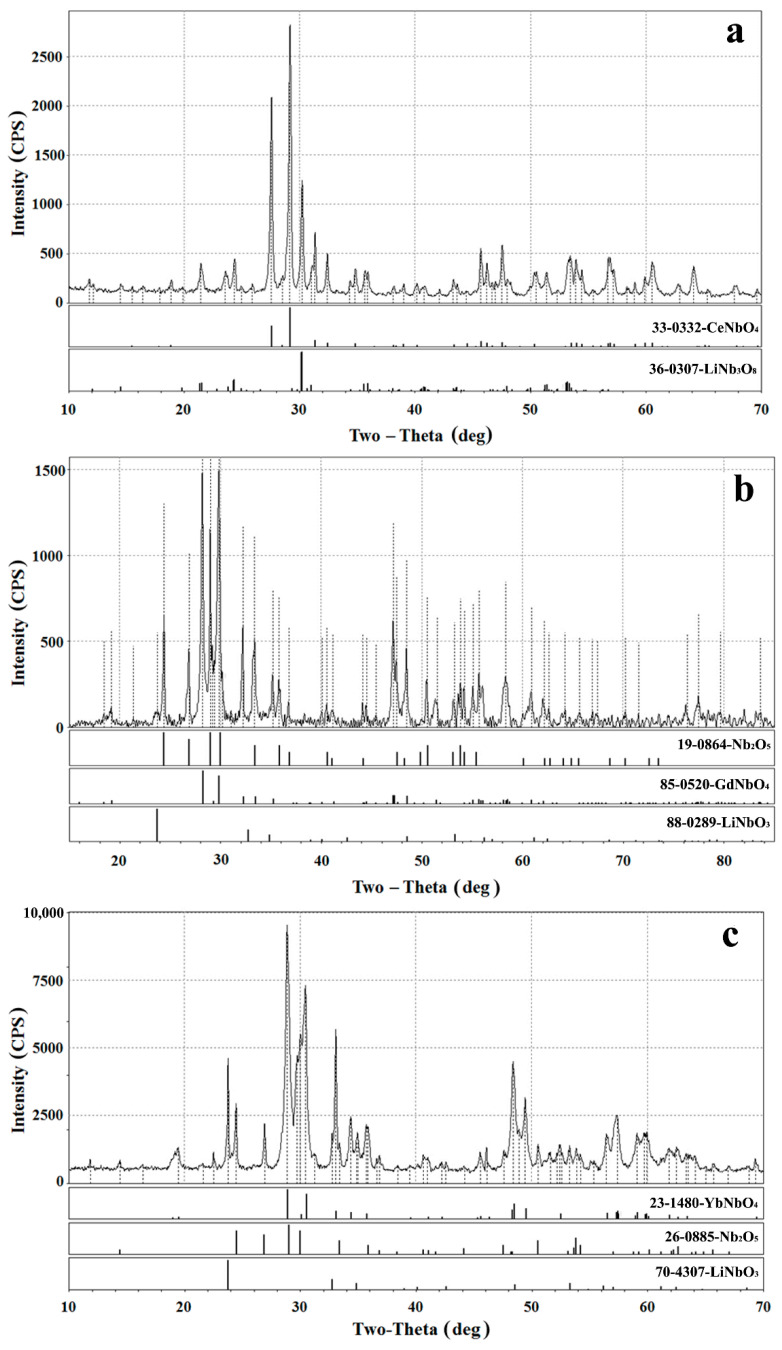
X-ray diffractograms of the products of the lithium ions heterovalent substitution reaction in LiNbO_3_ obtained after holding its powders in molten LiCl-LnCl_3_ mixture where Ln = Ce (**a**), Gd (**b**), Yb (**c**) under inert gas (argon) atmosphere at 700 °C.

**Figure 10 materials-15-03551-f010:**
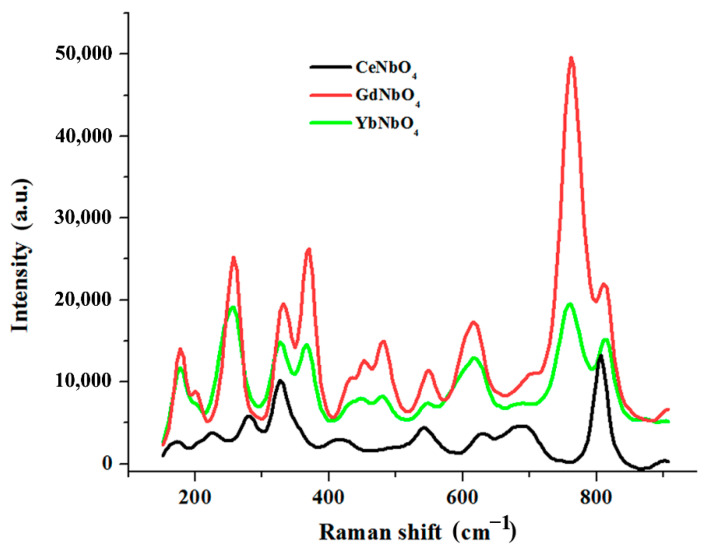
Raman spectra of the heterovalent ionic substitution reaction products formed by the interaction of the LiNbO_3_ powders with molten LiCl-LnCl_3_ (Ln = Ce, Gd, Yb) mixtures at 700 °C.

**Figure 11 materials-15-03551-f011:**
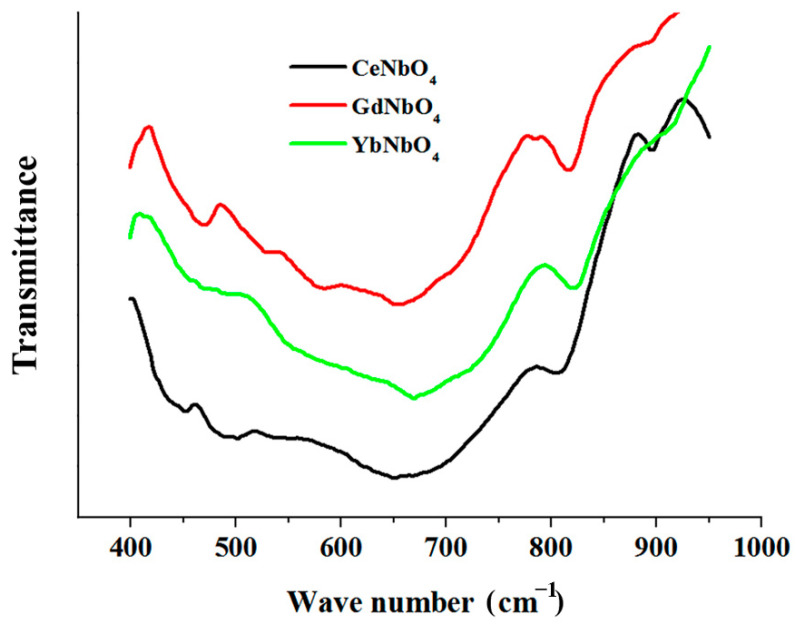
Infra-red spectra of the heterovalent ionic substitution reaction products formed by the interaction of the LiNbO_3_ powders with molten LiCl-LnCl_3_ (Ln = Ce, Gd, Yb) mixtures at 700 °C.

**Table 1 materials-15-03551-t001:** Parameters of vibrational bands in the Raman spectra of the initial and modified LiNbO_3_ single crystal samples.

FWHM (Full Width at Half Maximum)/cm^−1^		Normalized Intensity (I/I_874_)		Peak Maximum Position/cm^−1^	
Initial Sample	Modified Sample	Initial Sample	Modified Sample	Initial Sample	Modified Sample
24	24	0.37	0.59	237	237
32	34	0.41	0.60	269	268
34	36	0.40	0.50	327	325
33	33	0.19	0.24	370	370
33	34	0.53	0.56	430	431
24	26	0.05	0.08	477	479
35	36	0.60	0.78	583	583
29	37	0.12	0.37	620	621
112	92	0.06	0.09	649	659
40	40	1	1	874	874

**Table 2 materials-15-03551-t002:** Elemental composition of the reaction salt medium (molten KNO_3_-PbCl_2_ mixture) after the experiment.

[K]/mg∙dm^−3^	[Li]/mg∙dm^−3^	[Nb]/mg∙dm^−3^	[Pb]/mg∙dm^−3^
7522	0.035	˂0.001	1169

**Table 3 materials-15-03551-t003:** Elemental composition modified surface layer of the LiNbO_3_ single crystal from the results of the energy-dispersive X-ray spectroscopy (EDS).

[O]/at. %	[Nb]/at. %	[Pb]/at. %
72.0	27.0	1.0

## Data Availability

Data are contained within the article.

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
