# Peer review of "Molten Chlorides as the Precursors to Modify the Ionic Composition and Properties of LiNbO3 Single Crystal and Fine Powders"

_materials, 2022, doi:10.3390/ma15103551_

Round 1
Reviewer 1 Report
In the present study, the authors have tested the modification of LiNbO3 by molten salt, inclding molten CaCl2 and rare earth chlorides. Some issues should be addressed before publication:
(1) For the modification of single crystal LiNbO3, even with 1 mol% doping of Pb, a plunge in the peak intensity of LiNbO3 could be observed (Figure 4). How to explain this? It seems that the immersion has a negative effect on this single crystral.
(2) In the case of LiNbO3 in CaCl2-base chlorides salts or rare earth chloride salts, it seems it's becaues of chemical reaction (instead of substitution) that contributes to the formation of YbNbO4; that is, two or more different phases formed. Thus, the modification in this manusript refers to substitution or chemical reaction?
Reviewer 2 Report
Manuscript title: Molten Chlorides as the Precursors to Modify the Ionic Composition and Properties of the LiNbO3 Single Crystal and Fine Powders,
ID: materials-1709846
The manuscript focuses on the modification of the lithium niobate composition. The procedure follows the incorporation of molten salt mixtures containing calcium, lead, and rare-earth metals (REM) chlorides as the precursors. Subsequent structural, optoelectronic, and related specifications were assessed throughout this report. The manuscript might be considered for publication after the minor revision addressing the following points:
_All equations should be numbered in the manuscript.
_ figure 2 quality should be enhanced.
_ some sentences are hard to deliver the message, for instance:
“Using the last-named melts makes it possible to reduce the synthesis temperatures and ensure the more even distribution of the initial oxide particles involved in the reaction of the formation of the more complex compounds”. I suggest rephrasing.
_ check the grammar.
_ Kindly add parts in the introduction to highlight the significance of this work and add more recent close works.
_ Add more recent references 2021, 2020.
Best of luck
Reviewer 3 Report
The manuscript entitled “Molten Chlorides as the precursors to modify the ionic composition and properties of the LiNbO3 single crystal and fine powders” describes the synthesis of LiNbO3 crystals, their cationic modification, and the respective characterization of all materials. The comments and question related to this work are listed below:
The synthesis processes are innovative and rather interesting, and they are one of the contributions of this work.
In materials and methods section, figure 1. Please provide the PDF number used in the lithium niobate identification. Substitution of the word “pattern” for “Diffractogram” is also suggested
A deeper description of the chemical reactions to produce lithium niobate and its heterovalent substitutions is suggested.
Section 2.3. The text in paragraphs 150 through 157 should be deleted. In paragraph 137 “Diffraction pattern” could be replaced by “Diffractogram”, and so this in the rest of the text.
Results and discussion, section 3.1 figure 4. Crystal phases identification must be included and the corresponding PDF numbers
How thick is the PbNb2O6 film formed on the LiNbO3 crystal surface? Is it possible to measure it?
In page 7, paragraph 229, the authors concluded that the formation of a thin film on the LiNbO3 crystal surface was the result of an exchange reaction. Is the interphase between film and crystal well defined? Could some lead ions have entered in the LiNbO3 crystal structure as interstitial substituents? How this affect the properties of the crystal?
Additional structural analyses are recommended. Rietveld refinement is useful to know if some lead ions are entering into the LiNbO3 crystal structure and observations by means of a high-resolution electron microscope could be useful to analyze the microstructure of LiNbO3 crystals covered by the PbNb2O6 film.
Section 3.2, figure 6. Once again, the PDF numbers used in the crystalline phase identification must be provided. In the diffractogram displayed in this figure, the Bragg reflections about 21o and 25o in 2Θ seem to be unidentified. Is there any other crystalline phase in the sample?
In paragraph 281 through 286, the authors mentioned the formation of the crystalline phase CaNb2O6 and a solid substitution solution with cationic vacancies. According to authors, the obtaining of the latter is supported by the observed variations in the lattice parameters and the unit cell volume. Which Bragg reflections in the diffractogram were shifted because of this solution formation? How much calcium was entered into the LiNbO3 crystal structure?
How relevant is to know the positions of substituents ions and vacancies in the unit cell to understand the properties of the crystal?
A deeper structural analysis is proposed to understand the crystal structure of the solid solution Li(1-x) Ca (x/2) V(x/2) Li+NbO3. Rietveld refinement is highly suggested. Besides, observations using high resolution transmission electron microscopy could be useful to analyze the crystal structure of the solid solution. Selected area electron diffraction (SAED) or the obtaining of fast Fourier transform (FFT) can complete the structural analysis
Paragraph 297, the sentence is repeated.
Why the intensities of the Bragg reflections (figure 7) and the Raman signals (figure 8) are greater for the sample obtained at 700o C?
In section 3.3, figure 9 PDF numbers used in the crystal phase identification are required.
In figures 10 and 12, differences in Raman and IR spectra from sample to sample are necessary to be discussed in detail.
Reviewer 4 Report
The manuscript "Molten Chlorides as the Precursors to Modify the Ionic Composition and Properties of the LiNbO3 Single Crystal and Fine Powders" discusses the results of an experimental study of the full or partial heterovalent substitution of lithium ion in nanosized LiNbO3 powders and in the surface layer of LiNbO3 single crystal using molten salt mixtures containing calcium, lead, and rare-earth metals (REM) chlorides as the precursors. This article is interesting and can be accepted . The comments are below:
- It is a nice idea to mark the planes in the XRD data in Fig. 1.
- If possible, some digital image of the sample will be useful.
- A SEM micrograph will be helpful.
Round 2
Reviewer 3 Report
Thanks for answer all comments and questions